# Assessment of the Impact of Sensor-Based Ischemic Preconditioning with Different Cycling Periods on Upper Limb Strength in Bodybuilding Athletes

**DOI:** 10.3390/s24185943

**Published:** 2024-09-13

**Authors:** Xuehan Niu, Qifei Xia, Jie Xu, Li Tang

**Affiliations:** 1Physical Science College, Jishou University, Jishou 416000, China; gooutniuniu123@126.com; 2College of Aeronautics, Binzhou University, Binzhou 256600, China; 3Graduate School, University of Perpetual Help, Las Navas 6420, Philippines; 4School of Physical Education, Ankang University, Ankang 725000, China; xiaqifei_xaty1998@126.com; 5Institute of Sports Training, Xi’an Physical Education University, Xi’an 710064, China; 6College of Art, Wuhan Sports University, Wuhan 430079, China; xujie_whty@126.com

**Keywords:** ischemic preconditioning, muscle activation, upper limb strength, athletic performance enhancement, exercise training

## Abstract

**Objective:** This study designed experiments to explore the effects of ischemic preconditioning (IPC) intervention with different cycling periods on the upper limb strength performance of college male bodybuilding athletes. **Methods:** Ten bodybuilding athletes were recruited for a randomized, double-blind, crossover experimental study. All subjects first underwent pre-tests with two sets of exhaustive bench presses at 60% of their one-repetition maximum (1RM) to assess upper limb strength performance. They then experienced three different IPC intervention modes (T1: 1 × 5 min, T2: 2 × 5 min, T3: 3 × 5 min), as well as a non-IPC intervention mode (CON), followed by a retest of the bench press. An Enode pro device was used to record the barbell’s velocity during the bench press movement (peak velocity (PV), mean velocity (MV)); power (peak power (PP), mean power (MP)); and time under tension (TUT) to evaluate upper limb strength performance. **Results:** PV values: T1 showed significant increases compared to pre-tests in the first (*p* = 0.02) and second (*p* = 0.024) tests, and were significantly greater than the CON (*p* = 0.032); T2 showed a significant increase in PV in the first test (*p* = 0.035), with no significant differences in other groups. MV values: T1 showed a significant increase in MV in the first test compared to the pre-test (*p* = 0.045), with no significant differences in other groups. PP values: T1 showed a highly significant increase in PP in the first test compared to the pre-test (*p* = 0.001), and was significantly higher than the CON (*p* = 0.025). MP values: T1 showed highly significant increases in MP in both the first (*p* = 0.004) and second (*p* = 0.003) tests compared to the pre-test; T2 showed a highly significant increase in MP in the first test (*p* = 0.039) and a significant increase in the second test (*p* = 0.039). T1’s MP values were significantly higher than the CON in both tests; T2’s MP values were significantly higher than the CON in the first (*p* = 0.005) and second (*p* = 0.024) tests. TUT values: T1 showed highly significant increases in TUT in the first (*p* < 0.001) and second (*p* = 0.002) tests compared to the pre-test, and were significantly higher than the CON. **Conclusions:** (1) Single-cycle and double-cycle IPC interventions both significantly enhance upper limb strength performance, significantly improving the speed and power in exhaustive bench press tests, with the single-cycle IPC intervention being more effective than the double-cycle IPC intervention. (2) The triple-cycle IPC intervention does not improve the upper limb strength performance of bodybuilding athletes in exhaustive bench presses.

## 1. Introduction

Ischemic preconditioning (IPC), a procedure originally developed for the management of cardiovascular diseases, has been recognized for its potential in conditioning the heart against ischemic events. As one of the leading causes of global pathology, cardiovascular diseases, including acute myocardial infarction, present a sudden and often unanticipated challenge to heart health. The application of IPC as a preventive measure in these scenarios is well established. IPC, initially a medical procedure predominantly utilized in the treatment of cardiovascular diseases, has been successfully adapted to the realm of sports training as a conditioning method. It has become one of the pre-exercise conditioning activities (CAs) highly esteemed by coaches and trainers. As a non-invasive bio-regulatory technique, IPC serves as an intervention aimed at enhancing athletic performance without the need for invasive procedures. The intervention typically involves the application of brief occlusive pressure to the body’s muscles or tissue organs using inflatable cuffs. This brief ischemia is intended to stimulate the activation and spontaneous protective effect of organs and tissues through the phenomenon of reperfusion following the release of pressure [1]. Consequently, it aims to improve the athletic capabilities and muscle functions of the subjects in subsequent training or competitive events, including, but not limited to, explosive power, muscular endurance, and neuromuscular adaptation [2,3,4].

Fan Zihan [5] and Wang Zhou et al. [6] have successively conducted detailed analyses of the mechanisms by which IPC intervention enhances athletic performance. The results indicate that the mechanisms by which IPC intervention improves athletic performance primarily consist of the following three aspects: (1) IPC can effectively increase the oxygenation capacity of skeletal muscle and the blood flow of capillaries, significantly enhancing the muscle’s ability to utilize oxygen, and also markedly promoting mitochondrial biogenesis; (2) IPC can enhance the capacity for neural information transmission, thereby inducing a high degree of neuromuscular adaptation and increasing the level of muscle activation; (3) IPC can significantly increase the accumulation of a large number of endogenous nutrients, such as nitric oxide, opioid peptides, bradykinin, and adenosine secretion. These mechanisms interact with each other and collectively improve the muscle’s explosive power, endurance, and neuromuscular coordination, thereby enabling subjects to exhibit superior athletic performance in subsequent high-intensity training or competitive events. Currently, numerous scholars have explored the role of IPC in sports training from a methodological perspective, including aspects such as the site of occlusive intervention, the magnitude of occlusive pressure, the duration of IPC application, the cycling period of IPC, the differential effects of IPC usage, and its impact on athletic performance [7,8,9,10,11]. The performance-enhancing effects of IPC are primarily manifested in two areas: strength endurance and explosive power. In terms of strength endurance, studies have found that IPC intervention can significantly enhance the strength endurance performance of certain specialized athletes and fitness populations. For instance, Barbosa’s research team [12] arranged for healthy subjects to undergo lower limb remote IPC intervention with a cycle of 3 × 5 min at 220 mmHg, and found that the duration of the exhaustive grip strength test was prolonged by 11.2%. This study indicates that IPC intervention can significantly improve muscle endurance performance in resistance training. Regarding explosive power, Patterson et al. [13] arranged for healthy males to undergo bilateral lower limb IPC intervention with a cycle of 4 × 5 min at 220 mmHg, and observed that the peak power during the first, second, and third sprints in the bicycle final sprint test increased significantly by 2.4 ± 2.2%, 2.7%, and 3.7 ± 2.4%, respectively, showing a clear advantage over the control group. Kraus’s research team [14] further explored the impact of different IPC intervention methods on the enhancement of explosive power. This research team employed a randomized, double-blind, and crossover experimental design, conducting a controlled trial of unilateral alternating upper limb IPC intervention and bilateral upper limb IPC intervention, with the IPC intervention cycle consistently maintained at 4 × 5 min. The study found that both intervention methods significantly improved the peak power (increased by 3.2%) and mean power (increased by 2.7%) in the Wingate test. These results further confirm the potential of IPC intervention for enhancing endurance and explosive power, indicating that it can serve as an effective conditioning activity to help competitive athletes optimize and enhance their athletic performance.

Although the current research predominantly focuses on IPC interventions with multiple cycling periods, particularly the four-cycle pattern (4 × 5 min), an economic analysis of IPC intervention models reveals that traditional four-cycle or three-cycle IPC interventions require more than half an hour. This duration significantly increases the time and energy consumption for athletes during the warm-up phase. However, the true intent of conditioning activities is to rapidly activate the athletes’ physical functions and quickly achieve an economical state of athletic performance. Therefore, exploring IPC interventions with shorter time cycles is expected to become a major focus of future research. Against this backdrop, shorter IPC intervention cycles, such as single-cycle and double-cycle, have begun to receive widespread attention from scholars. Salagas’s research team [15] conducted an effective exploration of single-cycle IPC intervention. When it was arranged for 12 male subjects with resistance training experience to undergo single-cycle IPC intervention (occlusive pressure: 146.7 ± 15.0 mmHg, alternating pressure) in a self-controlled experiment, it was found that in subsequent bench press tests performed at 90% of 1RM for four sets of 12 s each, the IPC intervention could significantly increase the mean velocity (+9.0 ± 4.0%) and peak velocity (+7.8 ± 7.7%) during the bench press, and also significantly increase the number of repetitions (+7.6 ± 9.5%). This result not only confirms that single-cycle IPC intervention can significantly improve upper limb motor strength performance but also indicates significant improvements in strength endurance and explosive power. Another scholar [16] conducted an experimental study on double-cycle IPC intervention and found that double-cycle IPC intervention at an occlusive pressure of 220 mmHg could significantly increase the subjects’ jumping height (9.0 ± 9.1%), suggesting that the double-cycle IPC intervention model may have a positive impact on lower limb explosive power. Nevertheless, there is currently a lack of direct comparative studies on the impact of different cycling periods of IPC intervention on athletic performance.

IPC represents a novel non-invasive strategy that has garnered attention for its potential to augment athletic performance, particularly in the context of high-intensity resistance training. The rationale for selecting bodybuilding athletes as the subject population in this investigation is multifaceted. Bodybuilding athletes are characterized by their pursuit of muscular hypertrophy and strength, which involves repetitive exposure to strenuous resistance exercises that elicit significant myocellular stress and subsequent recovery processes. This physiological milieu provides a unique opportunity to scrutinize the effects of IPC on muscle performance enhancement and recovery kinetics. The vascularity inherent in individuals with a well-developed musculature, as observed in bodybuilding athletes, is considered a critical factor in IPC efficacy. The dense capillary network facilitates the ischemic stimulus and subsequent reperfusion, which are pivotal for initiating the purported adaptive responses. Mechanistically, IPC is posited to activate a suite of endogenous protective mechanisms. The ischemic insult triggers the release of humoral mediators, such as adenosine and bradykinin, which are implicated in the modulation of vascular tone and the initiation of angiogenic processes. The ensuing reperfusion phase is characterized by a surge in blood flow, which facilitates the restoration of oxygen and nutrient delivery to the exercised muscles, thereby potentially enhancing muscle force generation and attenuating the accumulation of metabolic byproducts associated with fatigue. By focusing our investigation on bodybuilding athletes, we aim to elucidate the intricate interplay between IPC and the physiological adaptations that underpin strength and power outcomes. This approach is expected to yield insights that are not only pertinent to the bodybuilding discipline but also extrapolatable to other athletic populations engaged in resistance-based training regimens.

In summary, for specific specialized training populations, whether IPC interventions of different training cycles will lead to differences in athletic performance in the same test still needs to be clarified through further experimental research. Such comparative studies are of significant practical importance for optimizing IPC intervention protocols to meet the needs of different athletes and to enhance the efficiency of sports training.

## 2. Objective and Methods

### 2.1. Object

This study was conducted in the Physical Fitness Room at Wuhan from October to November 2023. A randomized cross-control design was paired with a self-controlled approach. Recruitment focused on students from the institution, applying strict criteria to an initial group of 30, resulting in a final study group of 10 qualified participants. Table 1 outlines their basic information. To counteract inherent subjective bias, the true purpose of the experiment was withheld until the conclusion, with participants informed only of an aim to explore whether occlusive pressure stimulation before exercise can enhance upper limb motor performance. The decision to select male participants was to control for hormonal and physiological variations that could impact muscle activation and training outcomes, ensuring consistency in the physiological measurements.

Eligibility was based on the following: (1) at least five years of resistance training experience, (2) proficiency in bench press and being capable of performing a bench press weight at least 0.7 times their body weight and (3) no history of chronic diseases such as heart disease or hypertension that was not under control.

Exclusion criteria included the following: (1) Any history of musculoskeletal injuries of the upper limbs or chest that could affect the ability to perform bench press exercises. (2) The presence of cervical or lumbar spine diseases that could be exacerbated by performing bench press exercises. (3) The use of any substances and equipment that could affect muscle strength or performance, such as a Weightlifting Belt, Wrist Wraps and steroids or stimulants.

An exercise risk assessment was conducted, reviewing each participant’s physical activity history and administering the Physical Activity Readiness Questionnaire (PAR-Q+) to evaluate their physical condition and ensure the safety of the protocol. The exercise environment underwent a thorough assessment to meet safety standards. Participants were fully informed about the study’s aims, methodology, and potential risks before providing their informed consent. This study has obtained consent from all participants and complies with the Declaration of Helsinki. It was approved by the Ethics Committee of Zhengzhou University’s School of Basic Medical Sciences, with the reference number ZZUIRB2023-JCYXY0019.

### 2.2. Methods

#### 2.2.1. Experimental Design and Process

The experimental design of this study is meticulously structured into two principal phases: the preparatory phase and the formal testing phase. Commencing three days ahead of the formal testing, the preparatory phase involves key initiatives such as subject recruitment and the meticulous collection of basic information, including age, height, weight, years of training experience and the estimated maximum bench press weight (1RM). Subsequent to this information gathering, an 1RM test is meticulously administered to each subject, ensuring the precision and safety benchmarks for the forthcoming experiments are met.

On the day of the formal test, after a standardized warm-up and pre-test assessment, subjects serve as their own controls in a series of ischemic preconditioning (IPC) interventions {T1: single cycle (1 × 5 min), T2: double cycle (2 × 5 min), and T3: triple cycle (3 × 5 min)} and a non-IPC control intervention (CON), with the sequence determined by random assignment. The IPC intervention protocol adheres strictly to the methodology established by Rodrigues et al. [17], utilizing a uniform IPC occlusive pressure of 170 mmHg applied to the upper arm near the proximal end, with alternating intervention methods between limbs and varying only the cycle periods. A crucial enhancement to this protocol involves the integration of advanced sensor technology to monitor and record physiological responses during the IPC intervention. These sensors provide continuous, real-time data, ensuring the accuracy of the intervention and the subsequent analysis. CON receives a minimal occlusive pressure of 20 mmHg as a “sham” IPC intervention, establishing a placebo effect control group. A 48 h interval is mandated between different intervention modes to prevent cumulative effects that might skew the experimental data and to mitigate the risk of exercise-induced fatigue or injury.

Post the pre-test and following the IPC or sham IPC intervention, subjects undergo a standardized upper limb strength performance test. This test is designed in accordance with established IPC research protocols, utilizing the same testing methodology as the studies by Wilk [18] and Valenzuela [19]. The test involves two sets of exhaustive bench press exercises at 60% of 1RM, with a 2 min intermission between sets. Throughout the test, state-of-the-art sensors, such as the Enode pro power collection device (Simeier, Guangzhou, China), are employed to capture real-time velocity and power metrics of the barbell during the bench press, offering a precise reflection of the subjects’ upper limb strength performance under various IPC conditions. Refer to Figure 1 for a detailed visual representation of the experimental flowchart.

#### 2.2.2. Main Test and Observation Indicators

(1) Bench Press 1RM Test

Three days prior to the formal experiment, all participants were arranged by the testing staff of this experiment to undergo a 1RM bench press test to establish the maximum strength output of the participants in bench press training, providing an accurate load benchmark for the subsequent experimental design. The 1RM bench press test protocol adopted the same testing scheme as in previous studies [17,20]: Participants determined the order of the 1RM bench press test in a random and balanced manner (drawing lots) and verbally reported their estimated 1RM bench press values to the testers. All participants underwent the same standardized warm-up procedure during the test. They first performed a dynamic warm-up, with all participants walking on a treadmill at a speed of 5–6 km/h for 5 min and activating the shoulder and chest muscle groups during this process. Subsequently, participants underwent a specialized warm-up for the bench press, using loads of 20%, 40%, and 60% of the estimated 1RM, completing 15, 10, and 5 repetitions of the bench press, respectively, to adapt to the high-load 1RM bench press test that followed. Participants were required to perform the bench press with a unified standard movement: They were instructed to use a unified standard movement for the bench press to ensure the accuracy of the test. The barbell must touch the chest on the descent, and the elbows must be fully extended on the push, achieving the complete standard of the bench press movement. The testers used a metronome to control the participants’ rhythm, with the eccentric phase (Point A → Point B) lasting 2 s, and the concentric phase (Point B → Point C) being required to be completed at the fastest speed. After the specialized warm-up, participants performed stretching of the pectoralis major, deltoids, and triceps for 3 min to prevent muscle strain. The official 1RM test began at 80% of the estimated 1RM weight, gradually increasing the weight by 4–9 kg each time until the participant could not complete the specified number of repetitions. Between each weight increase, participants had a 2 min rest period. If the weight was successfully lifted in the attempt, it would continue to increase by 4–9 kg; if it failed, it would decrease by 2–4 kg. In this way, the participant’s 1RM was determined in 3–5 attempts. The 1RM of all participants in the bench press was determined in 5 experiments. Figure 2 is an illustrative diagram of the bench press 1RM.

(2) IPC Intervention

This study adopted the upper limb IPC intervention protocol designed by the research team of Rodrigues et al. [17]: each subject was required to use a uniform IPC occlusive pressure, intervention site, and method of intervention. The occlusive pressure was set at 170 mmHg, the intervention site was the upper arm close to the proximal end, and the intervention method was alternating between upper limbs. The experimental group varied only in the cycling period of the IPC intervention as the sole variable. In the single-cycle IPC intervention, subjects were first required to undergo 5 min of upper limb occlusive intervention to create an ischemic environment, followed by the release of occlusive pressure to promote blood reperfusion for another 5 min (starting with one arm, then alternating to the other arm for occlusion/release). The double-cycle IPC intervention, based on the single-cycle IPC intervention, repeated the cycle of pressurization and depressurization to ensure that each arm underwent two rounds of 5 min of pressurization and 5 min of depressurization and reperfusion. The triple-cycle IPC intervention required subjects to complete three rounds of 5 min of pressurization and 5 min of depressurization and reperfusion for each arm. Figure 3 illustrates the IPC intervention; Figure 4 displays the IPC equipment: the Theratools BFR device (Simeier, Guangzhou, Guangdong, China).

(3) Upper Limb Strength Test

The study employed an upper limb strength testing protocol consistent with previous research [18]: all subjects performed two sets of exhaustive bench press tests at 60% of their 1RM, with a 5 min interval between each set to ensure the adequate recovery of physical strength before the next test, preventing the occurrence of exercise fatigue that could affect the test results. The requirements for the bench press movement in the upper limb strength test were kept consistent with the 1RM test, with the only change being the speed of the bench press, which was modified to be pushed up as quickly as possible with individual maximum explosive power, meaning that both the eccentric and concentric phases of each repetition of the bench press were performed at the maximum possible speed, with exhaustion defined as the inability to complete the bench press with the standard movement. The Enode pro (Germany), a sports performance power data collection device, was used to record the time under tension (TUT), peak power output (PP), mean power output (MP), peak velocity (PV), and mean velocity (MV) of each set of bench presses during the test for subsequent data analysis. Figure 5 illustrates the power curve recorded during the upper limb strength test using the Enode pro; Figure 6 displays the Enode pro device. The Enode pro sensor was selected for its high precision and reliability, providing continuous and real-time data that ensure the accuracy of the intervention and the subsequent analysis. Through these measurements, we were able to conduct a detailed analysis of the effects of IPC intervention on muscle activation and athletic performance, which is crucial for understanding the role of IPC in enhancing the effectiveness of sports training [21].

(4) Experimental Control

In the present study, due to the use of a within-subjects experimental design, it is essential to exert strict control over potential confounding factors in the experimental process to ensure the accuracy and reliability of the results. The specific control measures are as follows: ① To monitor the changes in the upper limb strength performance of the subjects throughout the experimental testing period, although this study is an acute experiment, considering the four tests conducted by random drawing, subjects were required to avoid any specialized training related to bodybuilding during the experimental period. This included, but was not limited to, high-intensity resistance training, aerobic exercise, and physical fitness training. This measure is intended to prevent muscle damage and exercise fatigue caused by high-load resistance training, as well as to prevent exercise injuries that could lead to subjects withdrawing from the experiment. Additionally, this helps to avoid the “dynamic increase effect” of strength during the experiment, ensuring the accuracy of the 1RM data. ② The experiment was conducted in two stages: the bench press 1RM test and the formal test, with a 3-day interval between the two. This arrangement is to mitigate the exercise fatigue and potential exercise injuries that the maximum load 1RM test might cause, ensuring that the subjects are in the best physical condition during the formal test. ③ Throughout the experimental phase, subjects were required to strictly control their diet and daily routine to regulate their circadian rhythm. At the same time, the daily living habits of all subjects should be as consistent as possible to reduce interference from external factors. This includes a balanced diet, adequate hydration, sufficient and quality sleep, and other lifestyle habits that could affect the test results.

### 2.3. Statistical Analysis

After the completion of the experiment, the collected data were entered into Excel V2.5.294.2024 for the systematic categorization of all raw data, and the means and standard deviations were recorded. SPSS 25.0 was utilized for differential analysis, employing a repeated measures analysis of variance for the pre-experimental and post-experimental strength performance indicators within groups, and a one-way analysis of variance for the post-experimental strength performance indicators between groups. This analysis was conducted to assess the differences in the impact of IPC intervention with different cycling periods on the upper limb strength performance of college male bodybuilding athletes. In the testing process, a *p*-value of less than 0.05 indicates a level of significant difference, while a *p*-value of less than 0.01 indicates a level of highly significant difference.

## 3. Results

### 3.1. The Effect of IPC Intervention with Different Cycle Periods on Peak Velocity (PV)

The results shown in Table 2 and Figure 7 indicate that the within-group comparative analysis revealed the experimental group had significantly higher peak velocity (PV) values after single-cycle IPC intervention in both the first (*p* = 0.02) and second (*p* = 0.024) exhaustive bench press tests compared to the pre-tests (*p* < 0.05), with the PV value in the first bench press significantly greater than that of the CON (*p* = 0.032). After double-cycle IPC intervention, the experimental group also exhibited a significant increase in PV values in the first exhaustive bench press test (*p* = 0.035) compared to the pre-tests (*p* < 0.05), while no significant differences were observed in the other groups’ pre-and post-test comparisons (*p* > 0.05). Inter-group comparisons found no significant differences in post-test PV values among all groups (*p* > 0.05).

### 3.2. The Effect of IPC Intervention with Different Cycling Periods on Mean Velocity (MV)

The results shown in Table 3 and Figure 8 indicate that the within-group comparative analysis revealed that after single-cycle IPC intervention, the experimental group had a significantly higher mean velocity (MV) in the first exhaustive bench press test (*p* = 0.045) compared to the pre-test (*p* < 0.05), while no other groups showed significant differences between pre-and post-test values (*p* > 0.05). Inter-group comparisons found no significant differences in post-test MV values among all groups (*p* > 0.05).

### 3.3. The Impact of IPC Intervention with Different Cycling Periods on Peak Power (PP)

The results shown in Table 4 and Figure 9 indicate that the within-group comparative analysis revealed that after single-cycle IPC intervention, the experimental group exhibited a highly significant increase in peak power (PP) values in the first exhaustive bench press test (*p* = 0.001) compared to pre-test values (*p* < 0.01), while no other groups displayed significant differences between pre-and post-test values (*p* > 0.05). Inter-group comparisons found that the experimental group with single-cycle IPC intervention had significantly higher PP values in the first exhaustive bench press test (*p* = 0.025) compared to the CON, while no significant differences were observed among the post-test PP values of the other groups (*p* > 0.05).

### 3.4. The Impact of IPC Intervention with Different Cycling Periods on Mean Power (MP)

The results shown in Table 5 and Figure 10 indicate that the within-group comparative analysis revealed that after single-cycle IPC intervention, the experimental group had a highly significant increase in mean power (MP) values in both the first (*p* = 0.004) and second (*p* = 0.003) exhaustive bench press tests compared to pre-test values (*p* < 0.01). After double-cycle IPC intervention, the experimental group exhibited a highly significant increase in MP in the first exhaustive bench press test (*p* < 0.01) and a significant increase in the second test (*p* = 0.039). Inter-group comparisons found that the experimental group with a single-cycle IPC intervention had a highly significant increase in MP values in both the first (*p* = 0.002) and second (*p* < 0.001) exhaustive bench press tests compared to the CON (*p* < 0.01). The experimental group with a double-cycle IPC intervention had a highly significant increase in MP in the first exhaustive bench press test (*p* = 0.005) and a significant increase in the second test (*p* = 0.024) compared to the CON (*p* < 0.05), while no other groups showed significant differences in post-test MP values (*p* > 0.05).

### 3.5. The Impact of IPC Intervention with Different Cycling Periods on Time under Tension (TUT)

The results shown in Table 6 and Figure 11 indicate that the within-group comparative analysis revealed that after single-cycle IPC intervention, the experimental group had a highly significant increase in TUT values in both the first (*p* < 0.001) and second (*p* = 0.002) exhaustive bench press tests compared to pre-test values (*p* < 0.05), while no other groups showed significant differences between pre-and post-test values (*p* > 0.05). Inter-group comparisons found that the experimental group with a single-cycle IPC intervention had significantly higher TUT values in both the first (*p* = 0.029) and second (*p* = 0.015) exhaustive bench press tests compared to the CON, while no other groups showed significant differences in post-test TUT values (*p* > 0.05).

## 4. Discussion

### 4.1. The Impact of Single-Cycle IPC Intervention on Upper Limb Strength Performance

The results of this study indicate that after a single-cycle IPC intervention, all indicator values of the subjects in the first bench press were significantly higher than the pre-test, with the mean power (MP) and time under tension (TUT) values showing the most significant improvements, not only being highly significantly greater than the pre-test values but also highly significantly greater than the CON. In the second bench press, except for the peak power (PP) value, which showed no significant difference, the changes in other indicators were consistent with the first bench press. This study confirms that single-cycle IPC intervention can significantly enhance the upper limb strength performance of bodybuilders during bench press training, particularly in the significant enhancement of explosive power (MP value) and strength endurance (TUT value). The findings of this study support the results of previous research: Salagas’s research team [15], after arranging for 12 male subjects with resistance training experience to undergo single-cycle IPC intervention (occlusive pressure: 146.7 ± 15.0 mmHg, alternating pressure) in a self-controlled experiment, found that in subsequent bench press tests performed at 90% of 1RM for 4 sets of 12 s each, IPC intervention could significantly increase the mean velocity (+9.0 ± 4.0%) and peak velocity (+7.8 ± 7.7%) during bench press, as well as significantly increase the number of repetitions (+7.6 ± 9.5%). This study also revealed that single-cycle IPC intervention could significantly improve the upper limb sports strength performance of the subjects, manifesting in significant improvements in strength endurance and explosive power.

A synthesis of previous research and the findings of the present study reveals that single-cycle IPC intervention has a positive effect on upper limb strength performance. This enhancement can be attributed to the reduction of exercise-induced fatigue [15] and a significant increase in neuromuscular activation [22,23]. Salagas et al. [15], in their study of single-cycle IPC intervention, analyzed changes in the subjective fatigue index RPE after IPC intervention and noted a significant decrease in RPE values during the first set of bench presses in the IPC experimental group. This suggests that single-cycle IPC intervention may improve exercise performance in bench press training by reducing the sensation of muscle fatigue. Furthermore, the research team of de Oliveira Cruz [22,23] has successively pointed out in their series of studies on IPC’s promotion of exercise performance that intermittent pressure application during IPC intervention leads to a coordinated increase in the muscle performance and surface electromyographic activity of the target muscle group. This heightened neural activation may be due to a significant increase in the concentration of metabolic byproducts stimulated by IPC intervention, such as the substantial release of opioids, bradykinins, and adenosine, which may lead to an accumulation of an “overload” state. This accumulation, by stimulating the third- and fourth-order afferent nerve centers, triggers a feedback mechanism that promotes the recruitment of a large number of muscle fibers, thereby improving exercise performance and significantly increasing velocity and power during exhaustive bench presses.

However, it is noteworthy that in the second exhaustive bench press test, the PP value of the single-cycle IPC intervention group declined compared to the first performance, with no significant difference from the pre-intervention state. This may be related to the greater exercise-induced fatigue caused by the first exhaustive bench press training. Marocolo et al. [24], in their research on the training effects induced by IPC intervention, pointed out that the positive or negative effects of IPC intervention on exercise performance generally depend on the degree of muscle and mental fatigue. Since the exhaustive bench press training used in this study generates greater mechanical and metabolic stress, it may produce greater muscle fatigue compared to the fixed-load bench press in previous studies, leading to a weakening of the positive effects of IPC pretreatment during the second bench press and an inability to continue to enhance upper limb strength performance.

### 4.2. The Impact of Double-Cycle IPC Intervention on Upper Limb Strength Performance

The results of this study indicate that in the first bench press, the peak velocity (PV) value was significantly greater than the pre-test, and the mean power (MP) value was highly significantly greater than both the pre-test and the CON. In the second bench press, only the MP value remained significantly greater than the pre-test and the CON, with no significant changes observed in the other indicators. Compared to the changes in subjects after single-cycle IPC intervention, overall, the effectiveness of double-cycle IPC intervention in enhancing upper limb strength performance is not as pronounced as that of single-cycle IPC intervention. However, the double-cycle IPC intervention did show significant improvements in certain indicators, such as PV and MP, suggesting that it can also ameliorate bench press strength performance. Due to the scarcity of experimental studies on double-cycle IPC intervention, there are few studies on upper limb double-cycle IPC intervention that align with this study, making extensive comparisons difficult. Nevertheless, one study on lower limb double-cycle IPC intervention produced results similar to this study, demonstrating that double-cycle IPC intervention can enhance athletic performance: Beaven’s research team [16] found that after subjecting 14 subjects to lower limb double-cycle IPC intervention with an occlusive pressure of 220 mmHg, there was a significant increase in the jumping height (9.0 ± 9.1%) in the repeated jump test, confirming that the double-cycle IPC intervention pattern has a promotional effect on lower limb explosive power performance. The research team also explored whether the training benefits produced by IPC intervention have continuity: after performing the same repeated jump test 24 h after the test on the same day, it was found that compared to the non-IPC intervention, both acceleration and jumping height in the IPC experimental group increased significantly, indicating that the performance-enhancing effects of IPC intervention indeed have a delayed effect.

### 4.3. The Impact of Triple-Cycle IPC Intervention on Upper Limb Strength Performance

The findings of this study indicate that triple-cycle IPC intervention does not enhance the upper limb strength performance of bodybuilding athletes, with no significant differences observed in all indicators of upper limb strength performance, resulting in a negative outcome. A review of the current academic research on triple-cycle IPC intervention reveals considerable divergence in its effects on athletic performance. Some studies align with the results of this study, suggesting that triple-cycle IPC intervention does not improve performance and may even reduce it: Tanaka et al. [25] discovered that when healthy individuals underwent triple-cycle IPC intervention at 300 mmHg, no enhancement was observed in the subsequent 20% MVIC test of the quadriceps muscle of the right knee, confirming that triple-cycle IPC intervention with higher occlusive pressure may not improve endurance performance. Slysz et al. [26] further indicated in subsequent research that the use of triple-cycle IPC intervention could lead to a significant reduction of 3.5% in the exercise duration of subjects, demonstrating that triple-cycle IPC intervention might not only fail to enhance athletic performance but could also decrease it.

Conversely, some studies support the notion that triple-cycle IPC intervention can improve athletic performance. De Groot et al. [27] showed that when 15 healthy adult males with a certain training foundation underwent lower limb triple-cycle IPC intervention, and the effects of the IPC intervention on athletic performance were assessed using continuous measurements of power output, oxygen consumption, respiratory exchange ratio, heart rate, blood pressure, and blood lactate as outcome indicators, it was found that although IPC intervention did not significantly affect the body’s respiratory exchange ratio, maximum heart rate, blood pressure, and blood lactate, it did significantly increase the maximum oxygen uptake from 56.8 mL/kg/min to 58.4 mL/kg/min, an increase of 3% (*p* = 0.003), and the maximum power output significantly increased from 366 W to 372 W, an increase of 1.6%. This study reveals that triple-cycle IPC intervention may have certain enhancing effects on aerobic and anaerobic capacity. However, in conjunction with the results of this study, which focused on changes in speed and power indicators during exhaustive bench press training, triple-cycle IPC intervention may not significantly enhance upper limb performance in exhaustive bench press tests. This could be related to the use of exhaustive bench press testing in this study, which brings greater mechanical and metabolic stress compared to the test movements used in previous studies. The simultaneous implementation of triple-cycle IPC intervention may generate greater metabolic stress, and the resulting exercise fatigue could be greater than that produced by the fixed-load bench press used in previous studies, thus potentially neutralizing the performance-enhancing effects of IPC intervention and resulting in no significant changes in indicators. Additionally, the subjects included in this study were college bodybuilding athletes, who may exhibit competitive differences compared to the trained healthy population in previous studies, which could also affect the efficacy of triple-cycle IPC intervention.

### 4.4. Research Limitations

This study has several limitations: (1) Although the current academic community has a relatively mature understanding of the four-cycle IPC intervention, this study did not include the traditional four-cycle IPC intervention in its initial experimental design. However, the subjects of this study are bodybuilding athletes, and previous studies have shown heterogeneity, including differences in sports projects, testing protocols, and outcome indicators. Therefore, the absence of a four-cycle IPC intervention in this study presents certain limitations. It is currently unknown whether traditional four-cycle IPC intervention has a promoting effect on the athletic performance of bodybuilding athletes during exhaustive bench press training, and further research is needed to explore this. (2) The main outcome indicators of this study were speed, power, and time under tension in bench press movements, thus only analyzing the impact of different IPC intervention cycles on upper limb strength performance from the perspective of biomechanics. However, IPC intervention is closely related to neuromuscular adaptation, metabolic stress changes, and hemodynamic changes. Therefore, the impact of IPC intervention with different cycles on athletic performance should be explained from multiple perspectives, especially in terms of changes in blood flow velocity, metabolic substances in the blood, and electromyography changes in muscles, which require further analysis and verification. (3) This study adopted an acute experimental design; thus it can only effectively evaluate the IPC intervention on two dimensions of strength performance (explosive muscle strength and strength endurance), lacking an examination of the maximum muscle strength indicator. Therefore, long-term experimental research is needed in the future to explore whether IPC intervention can effectively improve maximum muscle strength. (4) Our study reveals the significant effects of specific IPC intervention cycles on the upper limb strength performance of bodybuilding athletes. Given the limitations of our sample size (n = 10), we acknowledge that this may affect the generalizability of the results. Future studies should expand the sample size to enhance the universality and statistical strength of the findings. Furthermore, as our study focuses on bodybuilding athletes, the results may not be directly applicable to athletes of other sports types. Therefore, we suggest that subsequent studies consider a broader range of athletic populations, including endurance athletes, to explore the effects of IPC under varying training loads and sports demands. Endurance athletes, due to their prolonged aerobic metabolic demands, may be particularly responsive to the blood reallocation and muscle endurance improvements induced by IPC. The existing research has indicated that IPC can enhance the efficiency of oxygen utilization in muscle tissue and promote endurance performance by augmenting mitochondrial function [4]. Thus, applying IPC to endurance athletes may reveal its potential to improve exercise efficiency and delay fatigue.

### 4.5. Originality and Contribution

In this study, we observed significant enhancements in upper limb strength performance following a single-cycle IPC (T1), corroborating the findings of Salagas et al., who reported increased mean and peak velocities during bench press exercises after single-cycle IPC in resistance-trained males. However, we noted that double-cycle IPC (T2), while also enhancing upper limb strength, did not achieve the same level of significance as the single-cycle IPC. This divergence from the results of Beaven et al., who found double-cycle IPC to significantly improve lower limb explosive power, may be attributed to the differential muscular and metabolic responses between upper and lower extremities, as well as distinct hemodynamic responses to IPC occlusion. Furthermore, our results indicated that triple-cycle IPC (T3) did not further augment upper limb strength performance, aligning with Tanaka et al., who observed no effect of triple-cycle IPC on quadriceps endurance. We hypothesize that the excessive metabolic stress and muscle fatigue induced by triple-cycle IPC might counteract the potential benefits of IPC preconditioning. Our findings contribute to the field of sports science by providing novel insights into the impact of varying IPC cycle durations on the upper limb strength performance of a specific athletic population—bodybuilding athletes. Our study emphasizes the necessity of optimizing IPC protocols to achieve the maximal enhancement of athletic performance. Lastly, our results point the way for future research to explore the comprehensive effects of different IPC cycles on neuromuscular adaptations, metabolic stress, and hemodynamic changes, and how these factors collectively influence improvements in athletic performance.

## 5. Conclusions

This investigation into the effects of IPC with varying cycling periods on the upper limb strength performance of collegiate male bodybuilding athletes revealed that single-cycle and double-cycle IPC interventions significantly enhance upper limb strength, as evidenced by an increased peak velocity, mean power, and time under tension during exhaustive bench press tests, with the single-cycle intervention being more effective. In contrast, the triple-cycle IPC intervention did not yield significant improvements in strength performance, suggesting a potential threshold beyond which additional IPC cycles do not confer additional benefits. These results underscore the importance of optimizing IPC protocols to cater to the specific needs of athletes and highlight the potential of single-cycle and double-cycle interventions as effective conditioning activities to augment training outcomes. Given the special needs of endurance athletes for muscle endurance and efficiency during prolonged exercise, we believe that IPC may have a positive impact on their performance by optimizing muscle oxygenation and metabolic efficiency.

## Figures and Tables

**Figure 1 sensors-24-05943-f001:**
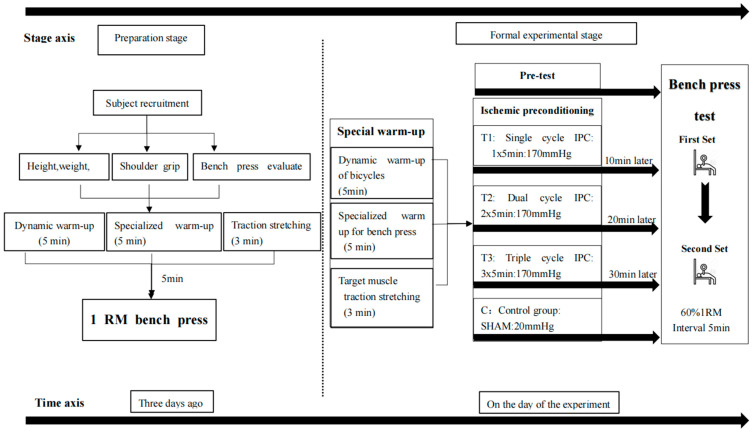
Experimental flowchart.

**Figure 2 sensors-24-05943-f002:**
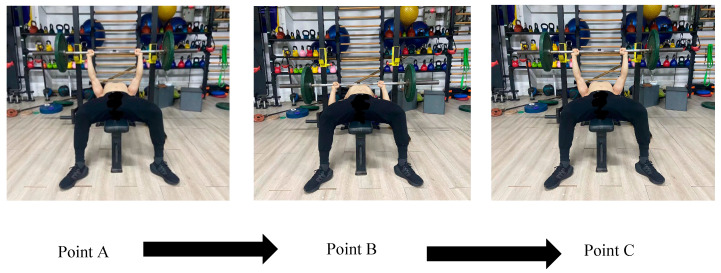
Upper limb 1RM strength test.

**Figure 3 sensors-24-05943-f003:**
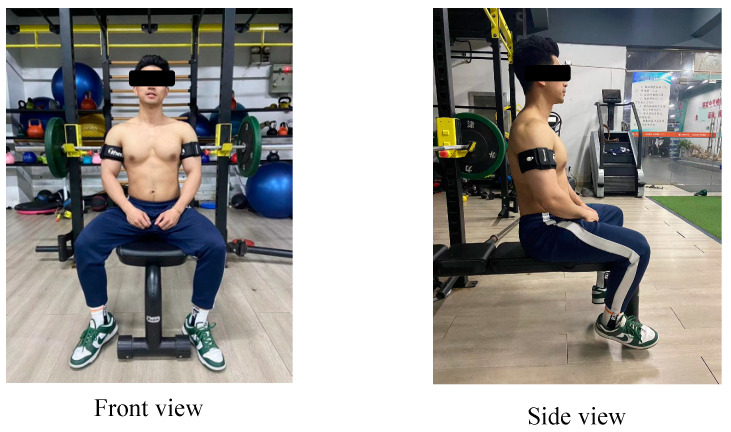
Schematic diagram of IPC intervention.

**Figure 4 sensors-24-05943-f004:**
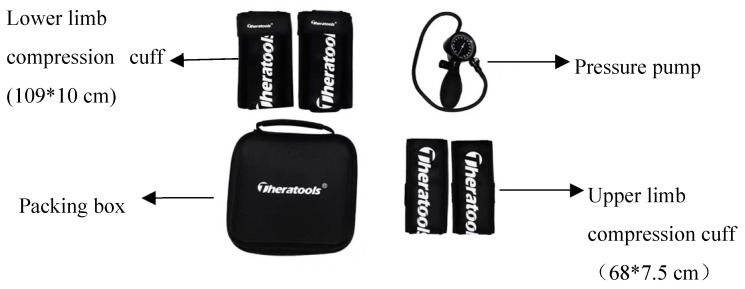
Pressurization equipment.

**Figure 5 sensors-24-05943-f005:**
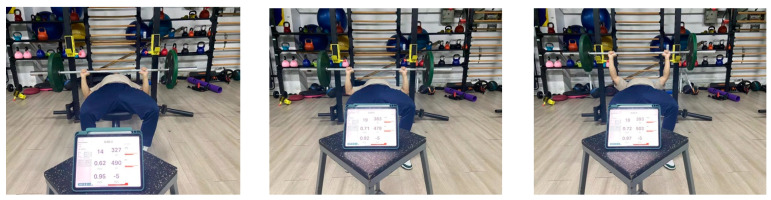
Schematic diagram of upper limb strength performance test.

**Figure 6 sensors-24-05943-f006:**
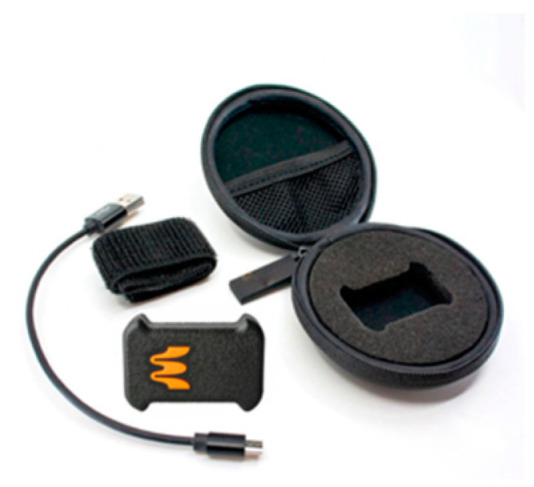
Enode pro.

**Figure 7 sensors-24-05943-f007:**
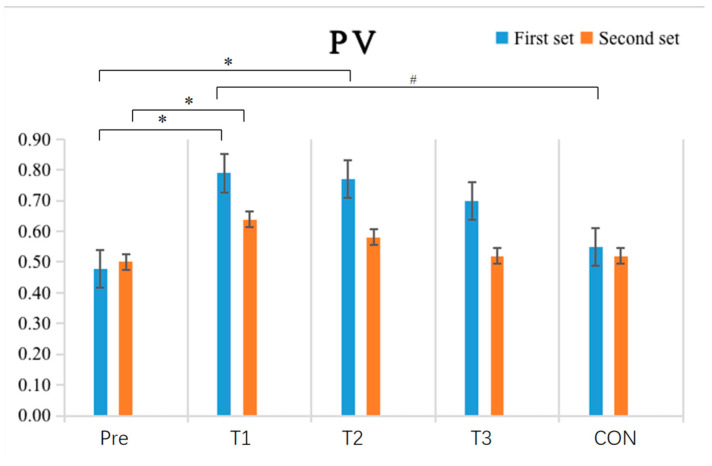
Schematic diagram of changes in upper limb strength performance PV index before and after the experiment (N = 10). Note: An asterisk (*) indicates a significant difference between pre-and post-tests within each group; a hash (#) indicates a significant difference compared to the T1.

**Figure 8 sensors-24-05943-f008:**
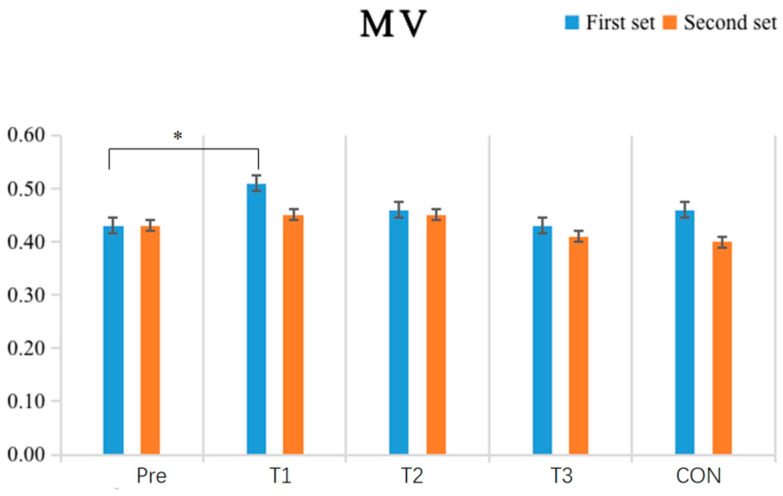
Schematic diagram of changes in the mean velocity (MV) indicator of upper limbsStrength performance before and after the experiment (N = 10). Note: An asterisk (*) indicates a significant difference between pre-and post-tests within each group.

**Figure 9 sensors-24-05943-f009:**
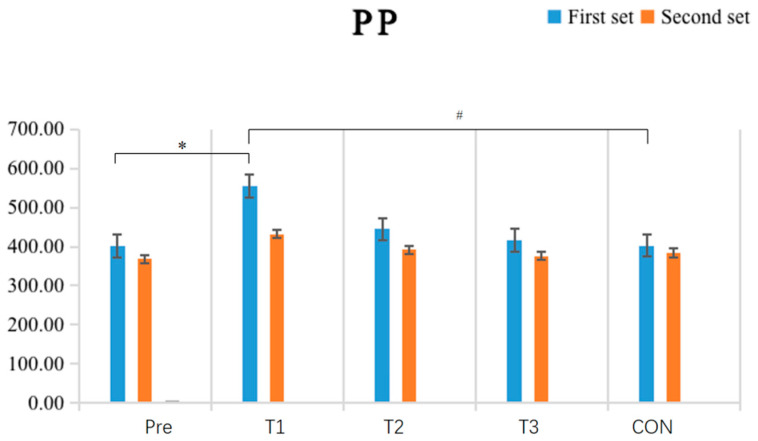
Schematic diagram of changes in the peak power (PP) indicator of upper limb strength performance before and after the experiment (N = 10). Note: An asterisk (*) indicates a significant difference between pre-and post-tests within each group; a hash (#) indicates a significant difference compared to the T1.

**Figure 10 sensors-24-05943-f010:**
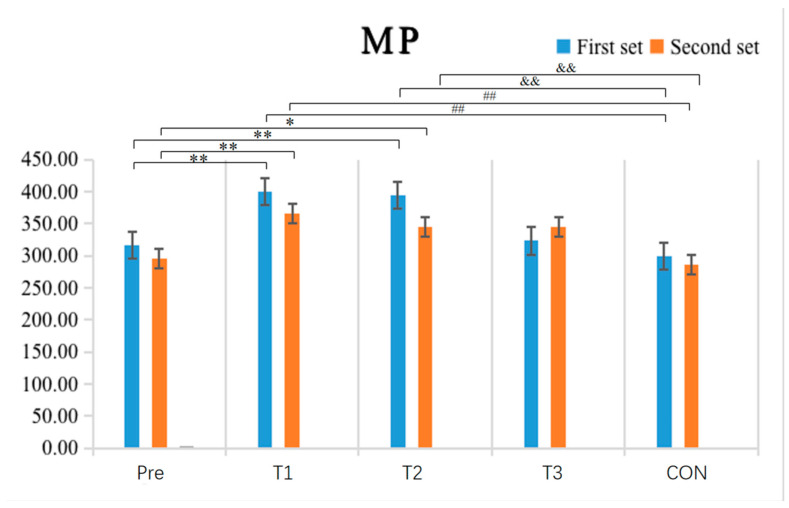
Schematic diagram of changes in the mean power (MP) indicator of upper limb strength performance before and after the experiment (N = 10). Note: An asterisk (*) denotes a significant difference between pre-and post-intervention measures within each group; a double asterisk (**) indicates a highly significant difference between pre-and post-tests within each group; a double hash (##) indicates a highly significant difference when compared to the single-cycle experimental group; a double ampersand (&&) indicates a highly significant difference when compared to T2.

**Figure 11 sensors-24-05943-f011:**
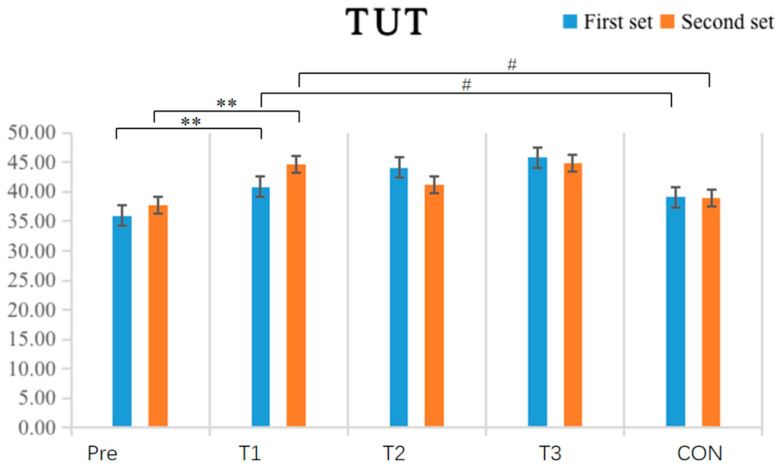
Schematic diagram of changes in the time under tension (TUT) indicator of upper limb strength performance before and after the experiment (N = 10). Note: a double asterisk (**) indicates a highly significant difference between pre-and post-tests within each group; a hash (#) indicates a significant difference compared to the T1.

**Table 1 sensors-24-05943-t001:** List of basic information of subjects (n = 10).

Year (Age)	Height (cm)	Weight (kg)	Training Years	Bench Press 1RM (kg)
20.34 ± 1.70	176.30 ± 6.17	80.15 ± 7.71	4.34 ± 2.11	124 ± 18.97

**Table 2 sensors-24-05943-t002:** Summary of changes in upper limb strength performance PV index before and after the Experiment (N = 10).

Bench Press Test	Group	Pre-Test (m/s)	Post-Test (m/s)	t-Value	*p*-Value	Time Effect	Group × Time Interaction	Effect Size (η^2^)
First	T1	0.476 ± 0.088	**0.821 ± 0.27 ***	−3.708	***p* = 0.02**	***p* < 0.001**	*p* = 1.667	0.442
	T2		**0.772 ± 0.252 ***	−3.506	***p* = 0.035**			
	T3		0.696 ± 0.291	−2.599	*p* = 0.186			
	CON		0.51 ± 0.061 **^#^**	−0.876	*p* = 0.986			
Second	T1	0.501 ± 0.118	**0.639 ± 0.186 ***	−2.59	***p* = 0.024**	***p* = 0.007**	*p* = 0.447	0.183
	T2		0.571 ± 0.038	−1.554	*p* = 0.773			
	T3		0.518 ± 0.103	−1.250	*p* = 0.911			
	CON		0.509 ± 0.104	−0.278	*p* = 0.994			

Note: An asterisk (*) indicates a significant difference between pre-and post-tests within each group; a hash (#) indicates a significant difference compared to the T1; a *p*-value less than 0.05 denotes a level of significance, while a *p*-value less than 0.01 denotes a level of high significance, with significant differences being presented in bold. (The same applies below).

**Table 3 sensors-24-05943-t003:** Summary of changes in upper limb strength performance MV index before and after the experiment (N = 10).

Bench Press Test	Group	Pre-Test (m/s)	Post-Test (m/s)	t-Value	*p*-Value	Time Effect	Group × Time Interaction	Effect Size (η^2^)
First	T1	0.44 ± 0.053	**0.509 ± 0.018 ***	−2.935	***p* = 0.045**	***p* = 0.017**	*p* = 0.249	0.148
	T2		0.456 ± 0.021	−0.948	*p* = 0.979			
	T3		0.421 ± 0.083	−0.102	*p* = 0.983			
	CON		0.458 ± 0.042	−1.02	*p* = 0.968			
Second	T1	0.42 ± 0.071	0.451 ± 0.05	−0.973	*p* = 0.773	*p* = 0.723	*p* = 0.199	0.004
	T2		0.466 ± 0.71	1.61	*p* = 0.741			
	T3		0.432 ± 0.23	0.901	*p* = 0.984			
	CON		0.399 ± 0.09	−0.823	*p* = 0.991			

Note: An asterisk (*) indicates a significant difference between pre-and post-tests within each group.

**Table 4 sensors-24-05943-t004:** Summary of changes in upper limb strength performance PP index before and after the experiment (N = 10).

Bench Press Test	Group	Pre-Test (m/s)	Post-Test (m/s)	t-Value	*p*-Value	Time Effect	Group × Time Interaction	Effect Size (η^2^)
First	T1	401.7 ± 58.91	**552.71 ± 78.4 ****	−4.584	***p* = 0.001**	***p* = 0.003**	***p* = 0.011**	**0.22**
	T2		421.3 ± 66.41	−1.295	*p* = 0.895			
	T3		415.67 ± 91.23	−0.419	*p* = 0.973			
	CON		403.93 ± 60.71 **^#^**	−0.067	*p* = 0.981			
Second	T1	361.3 ± 88.2	429.16 ± 183.1	−1.764	*p* = 0.647	*p* = 0.133	*p* = 0.706	0.062
	T2		391.59 ± 45.8	−0.652	*p* = 0.998			
	T3		354.13 ± 84.8	−0.24	*p* = 0.991			
	CON		383.1 ± 99.2	−0.419	*p* = 0.983			

Note: A double asterisk (**) indicates a highly significant difference between pre-and post-tests within each group; a hash (#) indicates a significant difference when compared to T1; a *p*-value less than 0.05 denotes a level of statistical significance, while a *p*-value less than 0.01 denotes a level of high statistical significance.

**Table 5 sensors-24-05943-t005:** Summary of changes in the mean power (MP) indicator of upper limb strength performance before and after the experiment (N = 10).

Bench Press Test	Group	Pre-Test (m/s)	Post-Test (m/s)	t-Value	*p*-Value	Time Effect	Group × Time Interaction	Effect Size (η^2^)
First	T1	309.21 ± 54.39	**391.61 ± 33.19 ****	−4.271	***p* = 0.004**	***p* = 0.001**	***p* < 0.001**	**0.409**
	T2		**394.92 ± 58.45 ****	−4.025	***p* = 0.004**			
	T3		307.11 ± 93.21	−0.382	*p* = 0.873			
	CON		299.68 ± 66.39 **^#&^**	1.646	*p* = 0.683			
Second	T1	296.01 ± 30.12	**359.2 ± 39.41 ****	−4.39	***p* = 0.003**	***p* < 0.001**	***p* = 0.003**	**0.311**
	T2		**345.02 ± 61.83 ***	−3.279	*p* = 0.039			
	T3		309.4 ± 39.16	−3.511	***p* = 0.074**			
	CON		286.33 ± 42.62 **^#&^**	0.833	*p* = 0.991			

Note: An asterisk (*) indicates a significant difference between pre-and post-tests within each group; a double asterisk (**) indicates a highly significant difference between pre-and post-tests within each group; a hash (#) indicates a significant difference compared to the T1; an ampersand (&) indicates a significant difference when compared to the double-cycle group; a *p*-value less than 0.05 denotes a level of statistical significance, while a *p*-value less than 0.01 denotes a level of high statistical significance.

**Table 6 sensors-24-05943-t006:** Summary of changes in the time under tension (TUT) indicator of upper limb strength performance before and after the experiment (N = 10).

Bench Press Test	Group	Pre-Test (m/s)	Post-Test (m/s)	t-Value	*p*-Value	Time Effect	Group × Time Interaction	Effect Size (η^2^)
First	T1	22.1 ± 4.9	**32.9 ± 6.51 ****	−4.767	***p* < 0.001**	***p* < 0.001**	***p* = 0.035**	**0.461**
	T2		29.5 ± 5.93	−3.105	*p* = 0.103			
	T3		28.8 ± 5.59	−2.799	*p* = 0.229			
	CON		24.1 ± 7.01 **^#^**	−0.437	*p* = 0.993			
Second	T1	17.31 ± 3.88	**26.4 ± 3.91 ****	−4.482	***p* = 0.002**	***p* = 0.007**	*p* = 0.447	**0.424**
	T2		23.1 ± 4.3	−3.004	*p* = 0.135			
	T3		20.8 ± 5.16	−1.871	*p* = 0.966			
	CON		18.9 ± 5.17 **^#^**	−0.936	*p* = 0.972			

Note: a double asterisk (**) indicates a highly significant difference between pre-and post-tests within each group; a hash (#) indicates a significant difference compared to T1; a *p*-value less than 0.05 indicates a level of statistical significance, while a *p*-value less than 0.01 indicates a level of high statistical significance.

## Data Availability

This study can provide data; please contact the first author (email gooutniuniu123@126.com).

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
