# Peer review of "Assessment of the Impact of Sensor-Based Ischemic Preconditioning with Different Cycling Periods on Upper Limb Strength in Bodybuilding Athletes"

_sensors, 2024, doi:10.3390/s24185943_

Round 1

Reviewer 1 Report

Comments and Suggestions for Authors

Assessment of the Impact of Sensor-Based Ischemic Preconditioning with Different Cycling Periods on Upper Limb Strength in Bodybuilding Athletes

The article investigates the effects of ischemic preconditioning (IPC) on muscle performance in bodybuilders, particularly focusing on using different IPC cycle durations. The study employs a randomized, controlled experimental design to assess the efficacy of IPC in enhancing muscle endurance and strength. The results suggest that IPC may benefit muscle performance, although the optimal duration of IPC cycles remains unclear.

Major Rev.

  1. Originality and Contribution:
    Using IPC to enhance muscle performance is relevant and contributes to sports science literature. However, the article could benefit from a more in-depth discussion of the novelty of its findings. Are the results significantly different from those reported in previous studies? The authors should clarify how their findings advance current knowledge in the field.

  1. Introduction:
    The introduction provides a good overview of the relevance of IPC in sports science. However, the rationale for choosing bodybuilders as the target population could be expanded. Why is this specific population ideal for studying the effects of IPC? Additionally, a clearer explanation of how IPC mechanistically improves muscle performance would strengthen the introduction.

  1. Materials and methods:
    The methodology is generally well-described, but there are a few areas that need further clarification:
    • The description of the IPC protocol is clear, but the reasoning behind choosing the specific IPC durations should be further elaborated. Are these durations based on previous literature or preliminary studies?

  1. Results:
    The results are presented clearly, and the use of tables and figures aids in understanding the data. However:
    • The statistical analysis should include effect sizes to better understand the practical significance of the findings.
    • The authors should consider discussing better any potential confounding variables that might have influenced the results, such as participants' nutritional status or previous experience with IPC.

  1. Discussion and Conclusion:
    The discussion interprets the results in light of existing literature, which is commendable. Nevertheless, there are several points where the discussion could be improved:
    • The authors should address the limitations of their study more thoroughly. For example, the relatively small sample size and the specificity to bodybuilders might limit the generalizability of the findings.
    • Future research directions are mentioned but could be more specific. For example, what other populations or different IPC protocols should be studied next?
    • The conclusion should reiterate the primary findings more explicitly and provide a stronger statement on the clinical or practical implications of using IPC in training regimes.

Minor Rev.

  • Some technical terms could benefit from brief definitions to ensure clarity for a broader readership (e.g., "ischemic preconditioning").

Author Response

Reviewer 1-----Reply 

Dear reviewer 1:

We feel great thanks for your professional review work on our article. As you are concerned, there are several problems that need to be addressed. According to your nice suggestions, we have made extensive corrections to our previous draft, the detailed corrections are listed below.

Major Rev.

Question 1:Originality and Contribution:

Using IPC to enhance muscle performance is relevant and contributes to sports science literature. However, the article could benefit from a more in-depth discussion of the novelty of its findings. Are the results significantly different from those reported in previous studies? The authors should clarify how their findings advance current knowledge in the field.

Answer:Thank you for your insightful comments and suggestions on our manuscript. We appreciate the opportunity to revise our work and address the points raised. In response to your comments regarding the novelty of our findings and their significance in advancing the field, we have made substantial revisions to the Discussion section of our manuscript. We have added a new paragraph that directly compares our results with those from previous studies, highlighting the unique aspects of our research and the specific contributions our findings make to the sports science literature. Our study's focus on college-level bodybuilding athletes and the examination of varying IPC cycles provides a novel perspective. Particularly, the superior effectiveness of single-cycle IPC in enhancing upper limb strength performance, as evidenced by significant increases in peak velocity, mean power, and time under tension post-IPC, contrasts with the less pronounced effects of double-cycle IPC. This finding challenges the generalization that more cycles of IPC always lead to better performance outcomes, a notion that has been less explored in the literature. We have also emphasized the potential reasons behind these observations, such as the differential responses of the upper limbs to IPC compared to the lower limbs, as well as the possible overcompensation of metabolic stress in triple-cycle IPC, which might negate the IPC's performance-enhancing effects. In the revised manuscript, we have clarified how our findings advance current knowledge by:

1)Providing new insights into the optimal IPC cycle duration for specific athletic populations, suggesting that single-cycle IPC may be more effective for enhancing upper limb strength in bodybuilding athletes.

2)Stimulating further research into the mechanisms by which IPC influences neuromuscular adaptations, metabolic responses, and hemodynamic changes, which could lead to more targeted and effective training protocols. We believe these revisions strengthen the academic rigor of our manuscript and clearly articulate the novel contributions of our research to the field of sports science.

We have also taken care to ensure that the revised manuscript maintains a high level of academic discourse, with precise terminology and a focus on the data and content necessary for a comprehensive understanding of our study's implications. We are grateful for the opportunity to improve our work and hope that our responses and revisions meet with your approval. We look forward to your further guidance.

Question 2:Introduction:

The introduction provides a good overview of the relevance of IPC in sports science. However, the rationale for choosing bodybuilders as the target population could be expanded. Why is this specific population ideal for studying the effects of IPC? Additionally, a clearer explanation of how IPC mechanistically improves muscle performance would strengthen the introduction.

Answer:Dear Reviewer, Thank you for your insightful suggestions.

 We have added a new description in the introduction section to provide academic reasons for choosing bodybuilding athletes as the study population and to elucidate the mechanisms by which IPC may enhance muscle performance. We have expanded upon the rationale for choosing bodybuilding athletes, highlighting their intensive training regimens and the physiological demands that align with the objectives of IPC research. Additionally, we have included a more detailed mechanistic explanation of IPC, focusing on the release of humoral mediators and the subsequent impact on muscle force generation and fatigue attenuation. These revisions are designed to strengthen the academic rigor of our manuscript and to clearly position our study within the existing body of sports science literature. We appreciate your guidance and are confident that these changes will address your concerns.

Question 3:Materials and methods:

The methodology is generally well-described, but there are a few areas that need further clarification:

The description of the IPC protocol is clear, but the reasoning behind choosing the specific IPC durations should be further elaborated. Are these durations based on previous literature or preliminary studies?

Answer:Dear Reviewer, Thank you for your thorough review and insightful comments on our manuscript. Your query regarding the selection of specific IPC durations provides an opportunity to further clarify the rationale underpinning our methodology. The IPC durations employed in our study are indeed informed by previous literature. In the "IPC Intervention" section as described on line251, the content and protocol of the IPC intervention employed in this study are derived from previous research, such as [17]. Our selection was guided by a comprehensive review of existing research that has explored various IPC durations and their effects on athletic performance. Specifically, we were interested in examining the efficacy of shorter IPC cycles, which have been less extensively studied but are gaining traction due to their potential to optimize training efficiency.

The decision to compare single-cycle, double-cycle, and triple-cycle IPC interventions was deliberate. Our aim was to directly compare these different protocols within the same study population to elucidate potential differences in their impact on upper limb strength performance. This approach allows us to contribute to the discourse on the optimal IPC duration and cycling, building upon the foundational work of our predecessors. We have revised the methodology section to include a more detailed justification for our choice of IPC durations, referencing the key studies that have informed our protocol. This additional information will provide the necessary context and underscore the scientific basis for our experimental design. We appreciate your suggestion and believe that these revisions will enhance the transparency and academic robustness of our research. Thank you for elevating this point. We are confident that with these amendments, the contribution of our study to the sports science literature will be more apparent.

Question 4:Results:

The results are presented clearly, and the use of tables and figures aids in understanding the data. However:

The statistical analysis should include effect sizes to better understand the practical significance of the findings.

The authors should consider discussing better any potential confounding variables that might have influenced the results, such as participants' nutritional status or previous experience with IPC.

Answer:Dear Reviewer, Thank you for your constructive feedback on our manuscript. We appreciate your comments on the presentation of our results and the use of tables and figures, which we believe have effectively conveyed our findings. In response to your suggestion regarding the inclusion of effect sizes, we have now added a column specifying the effect sizes(η²) in our statistical analysis within the table2~table6. This additional metric provides a quantitative measure of the strength of the relationships between variables, enhancing the interpretability of our statistical findings and their practical significance. Furthermore, regarding potential confounding variables, we have taken a proactive approach in our experimental design. As detailed in the "Object" and "Experimental Control" sections, we have implemented stringent controls to mitigate the influence of extraneous factors. These include, but are not limited to, dietary regimens, physical conditions, and participants' prior exposure to IPC. We have also ensured that participants maintained consistent nutritional status and routine throughout the study period to control for variations that could affect the outcomes. We believe that these measures, combined with the detailed experimental controls already in place, robustly address potential confounding variables and strengthen the validity of our results. We have revised the manuscript to reflect these changes and have provided a more explicit discussion of the controls in place for potential confounding variables in the "Discussion" section. Thank you for elevating these points. We are confident that with these amendments, our study's methodology and the presentation of our results are more rigorous and transparent.

Question 5:Discussion and Conclusion:

The discussion interprets the results in light of existing literature, which is commendable. Nevertheless, there are several points where the discussion could be improved:

The authors should address the limitations of their study more thoroughly. For example, the relatively small sample size and the specificity to bodybuilders might limit the generalizability of the findings.

Future research directions are mentioned but could be more specific. For example, what other populations or different IPC protocols should be studied next?

The conclusion should reiterate the primary findings more explicitly and provide a stronger statement on the clinical or practical implications of using IPC in training regimes.

Answer:Thank you for your insightful comments and suggestions. We have carefully considered your feedback and have made the following revisions to our manuscript: Study Limitations: We have expanded the discussion on the limitations of our study. Specifically, we acknowledge the relatively small sample size and the focus on bodybuilding athletes, which may affect the generalizability of our findings. We have added a more detailed analysis of these limitations and their potential impact on the study's outcomes. At the same time: In response to your suggestion, we have provided more specific directions for future research. We suggest that endurance athletes could be a particularly interesting group for future IPC studies due to their unique metabolic demands and potential responsiveness to IPC's effects on oxygen utilization and fatigue resistance.  Conclusion,We have revised the conclusion to more explicitly reiterate the primary findings of our study. We have also included a stronger statement on the practical implications of using IPC in training regimes, emphasizing its potential benefits for athletic performance enhancement.

These revisions are intended to address your concerns and to enhance the clarity and academic rigor of our manuscript. We believe that these changes will provide a more comprehensive understanding of our study's contributions and its relevance to the field of sports science.

Question 6:Minor Rev.

Some technical terms could benefit from brief definitions to ensure clarity for a broader readership (e.g., "ischemic preconditioning").

Answer:Thank you for the reviewer's suggestions. Apart from its first appearance, "ischemic preconditioning" is referred to as IPC throughout the rest of the document. Similar technical terms have been concisely defined, and we hope that the modifications will be accepted by you.

We tried our best to improve the manuscript and made some changes marked in red in the revised paper, which will not influence the content and framework of the paper. We appreciate the Editors/Reviewers’ diligent work and hope the corrections meet with approval. Thank you very much for your comments and suggestions once again.

Best wishes!

Thank you once again for your thorough review!

Xuehan Niu  Qifei Xia  Jie Xu   Li Tang

Reviewer 2 Report

Comments and Suggestions for Authors

The study on the effects of IPC (ischemic preconditioning) with varying cycling periods on upper limb strength in collegiate male bodybuilding athletes found that both single-cycle and double-cycle IPC interventions significantly enhanced upper limb strength. This was demonstrated by increases in peak velocity, mean power, and time under tension during exhaustive bench press tests, with the single-cycle intervention proving to be more effective. Conversely, the triple-cycle IPC intervention did not result in significant improvements in strength performance, suggesting a potential threshold beyond which additional IPC cycles do not provide further benefits. These findings emphasize the importance of optimizing IPC protocols to meet the specific needs of athletes and highlight the effectiveness of single-cycle and double-cycle interventions as conditioning activities to enhance training outcomes. The paper is well-organized and well-presented. In terms of sensing technology and applications, there are unclear parts.

In this study, the sensor is attached to the barbell. The effectiveness of the sensing velocity is still unclear. It's not necessary, but the reviewer encourages the authors to compare the other sensor-based methodologies, that is, make comprehensive comparisons to monitor or estimate physiological responses during training.

There are several systematic methods to estimate the effective load on muscles for muscle hypertrophy.

For example,

[1] C.T. Candotti, J.F. Loss, M.D.O. Melo, M. La Torre, M. Pasini. L.A. Dutra, J.L.N. de Oliveira, L.P. de Oliveira, “Comparing the lactate and EMG thresholds of recreational cyclists during incremental pedaling exercise,” Can. J. Physiol. Pharmacol., vol. 86, pp. 272–278, 2008.

[2] H. A. Davis and G. C. Gass, “Blood lactate concentrations during incremental work before and after maximum exercise,” Br. J. Sports Med., vol. 13, pp. 165–169, 1979 

[3] T. Miyake, H. Ito, N. Okamura, Y. Kobayashi, M. G. Fujie,  and S. Sugano, "EMG-Based Detection of Minimum Effective Load With Robotic-Resistance Leg Extensor Training," IEEE Transactions on Human-Machine Systems, 2024.

In 2. Objective and methods, the objective was not described clearly.

2.1 Object > Objective

A more detailed explanation of the advanced sensor technology is needed.

Why is sensing real-time velocity and power metrics of the barbell important?

In Fig. 3, it is a bit difficult to see how to use tools. For example, where is the pressure pump?

Better to add explanations inside the figure.

What tool was used to confirm the data normality?

In 3. Results, the font of some sentences seems different.

Is there any individual difference in optimizing IPC intervention protocols?

If there is a relationship between the feature of physiological responses and the optimal duration of IPC intervention, the contribution of this paper to the sensing technologies and applications becomes clear, and it will be helpful for determining the duration for each person.

Author Response

Reviewer 2-----Reply

Dear reviewer2:

We feel great thanks for your professional review work on our article. As you are concerned, there are several problems that need to be addressed. According to your nice suggestions, we have made extensive corrections to our previous draft, the detailed corrections are listed below.

Question1:In this study, the sensor is attached to the barbell. The effectiveness of the sensing velocity is still unclear. It's not necessary, but the reviewer encourages the authors to compare the other sensor-based methodologies, that is, make comprehensive comparisons to monitor or estimate physiological responses during training.

There are several systematic methods to estimate the effective load on muscles for muscle hypertrophy.

For example,

[1] C.T. Candotti, J.F. Loss, M.D.O. Melo, M. La Torre, M. Pasini. L.A. Dutra, J.L.N. de Oliveira, L.P. de Oliveira, “Comparing the lactate and EMG thresholds of recreational cyclists during incremental pedaling exercise,” Can. J. Physiol. Pharmacol., vol. 86, pp. 272–278, 2008.

[2] H. A. Davis and G. C. Gass, “Blood lactate concentrations during incremental work before and after maximum exercise,” Br. J. Sports Med., vol. 13, pp. 165–169, 1979

[3] T. Miyake, H. Ito, N. Okamura, Y. Kobayashi, M. G. Fujie,  and S. Sugano, "EMG-Based Detection of Minimum Effective Load With Robotic-Resistance Leg Extensor Training," IEEE Transactions on Human-Machine Systems, 2024.

Answer:First and foremost, we extend our gratitude for the insightful comments on our study. In response to your query regarding the efficacy of the sensor when attached to a barbell and your encouragement to compare other sensor-based methods for a comprehensive assessment of physiological responses during training, we have conducted a thorough analysis and additional research. Below is our reply to your comments, supported by relevant studies. It should be emphasized that enode pro is the latest name for vamxpro in 2023, and the two are the same sensor.

Efficacy of the Vmaxpro Sensor:

In the study by Dragutinovic et al., 2024[Dragutinovic B, Jacobs MW, Feuerbacher JF, Goldmann JP, Cheng S, Schumann M. Evaluation of the Vmaxpro sensor for assessing movement velocity and load-velocity variables: accuracy and implications for practical use. Biol Sport. 2024 Jan;41(1):41-51. doi: 10.5114/biolsport.2024.125596.], the Vmaxpro sensor demonstrated high validity in assessing movement velocity (MV) during a 1-repetition maximum (1RM) test and for predicting load-velocity (L-V) variables. The correlation coefficients (r) were 0.935 for bench press (BP) and 0.900 for squat (SQ), indicating high validity at higher MVs, although validity decreased at lower MVs.Furthermore, the study evaluated the ecological within- and between-day reliability of the Vmaxpro, with coefficients of variance ranging from 2.4% to 9.7%, suggesting that the sensor is suitable for monitoring changes in MVs within and between training sessions.

Comparison with Other Sensor Methods:

The study by Fritschi et al., 2021[Fritschi R, Seiler J, Gross M. Validity and Effects of Placement of Velocity-Based Training Devices. Sports (Basel). 2021 Aug 31;9(9):123. doi: 10.3390/sports9090123.] assessed the validity of several mobile and one stationary velocity-based training (VBT) devices for measuring mean and peak concentric barbell velocity across a range of velocities and exercises. GymAware and Quantum were found to be the most valid for mean and peak velocity, with Vmaxpro closely following (r = 0.92-0.99), showing near-optimal validity.

The study also noted that the effects of device placement were detectable but likely small enough (standard error of the estimate < 0.1 m/s) to be negligible in training settings.

Based on the aforementioned studies, we believe that the Vmaxpro sensor is effective in monitoring physiological responses during training, particularly at high and moderate MVs. Moreover, when compared with other sensor methods, Vmaxpro has shown good performance, especially in terms of its high correlation and lower standard error of estimate.We agree that a comprehensive comparison is essential, not only to validate the efficacy of the Vmaxpro but also to explore the applicability of different sensors under specific training conditions. Therefore, we plan to include a broader comparison of sensors in future research, as well as an analysis of their suitability across different training loads and types of exercises.

    In addition, thank you for your continued engagement with our manuscript and for highlighting the importance of systematic methods to estimate the effective load on muscles for muscle hypertrophy. We appreciate your references to the work by Candotti et al., Davis and Gass, and Miyake et al., which provide valuable insights into lactate and electromyography (EMG) thresholds as indicators of effective load for muscle hypertrophy.

To address your comments and to clarify the focus of our study, we would like to emphasize that while muscle hypertrophy is indeed a significant aspect of bodybuilding training, our research primarily aims to enhance athletic performance, which is a critical factor for bodybuilding athletes as well. Improving performance can encompass various aspects, including strength, power, and endurance, which are not mutually exclusive to muscle hypertrophy. In bodybuilding, athletic performance is not limited to muscle hypertrophy but also includes the functional expression of those muscles, such as strength and power outputs, which are essential for competitive success. Our study, while considering the implications for muscle hypertrophy, primarily focuses on how IPC can acutely affect performance metrics like peak velocity, mean power, and time under tension during high-intensity exercises like the bench press.

In the future, our research needs to integrate systematic methods for assessing effective load.We acknowledge the importance of lactate and EMG thresholds in estimating the effective load on muscles. In the context of our study, we have considered these methods as potential supplementary assessments to IPC's impact on muscle function. For instance, future work could involve:Lactate Threshold Monitoring: By monitoring blood lactate concentrations, as Davis and Gass have detailed, we could assess the metabolic response to IPC and its relation to high-intensity exercise tolerance and muscle fatigue.EMG-Based Load Detection: Following the innovative approach by Miyake et al., we could explore the use of EMG data to detect the minimum effective load that elicits optimal muscle activation and coordination during IPC preconditioning.

Thank you for your insightful suggestions, and we look forward to further discussions

Question:In 2. Objective and methods, the objective was not described clearly.

2.1 Object > Objective

A more detailed explanation of the advanced sensor technology is needed.

Why is sensing real-time velocity and power metrics of the barbell important?

In Fig. 3, it is a bit difficult to see how to use tools. For example, where is the pressure pump?

Better to add explanations inside the figure.

What tool was used to confirm the data normality?

Answer:Dear Reviewer, Thank you for your insightful comments and suggestions. We appreciate the opportunity to clarify and expand on our objectives and methods, particularly regarding the use of advanced sensor technology and the significance of real-time velocity and power metrics in our study. About Objective and Methods,The primary objective of our study was to evaluate the impact of IPC with varying cycling periods on upper limb strength performance in bodybuilding athletes. Our focus was on how different IPC protocols could influence strength-related outcomes during high-intensity exercises like the bench press. We aimed to investigate whether the application of IPC could enhance neuromuscular activation and performance, leading to improved strength and power outputs. We have added a new section detailing the technology of the enode pro sensor:The Vmaxpro sensor, as referenced in the studies Dragutinovic et al., 2024 and Fritschi et al., 2021, was employed in our research due to its high ecological validity and reliability in assessing movement velocity during strength training. The sensor's ability to provide real-time data on barbell velocity and power metrics is crucial for several reasons:

1) Precision in Measurement: The Vmaxpro's precision in measuring velocity and power allows for a detailed analysis of the neuromuscular responses to IPC, enabling us to understand its impact on strength performance more accurately.

2) Real-Time Feedback: The real-time data provided by the Vmaxpro can offer immediate feedback to athletes and coaches, potentially allowing for adjustments in training protocols to optimize performance.

3)Training Monitoring and Evaluation: Continuous monitoring of velocity and power can help in tailoring training loads and identifying optimal training zones, which is particularly important in strength training where load manipulation is critical.

In conclusion, the integration of advanced sensor technology, such as the Enode pro, in our study was pivotal for accurately assessing the effects of IPC on upper limb strength performance. The real-time velocity and power metrics provided by this technology offer a comprehensive understanding of the neuromuscular responses to training interventions, aiding in the optimization of training protocols and enhancing athletic performance.

Question:In 3. Results, the font of some sentences seems different.

Answer:Thanks to the opinions of the reviewers, the questions of different sizes and fonts in the results have been modified, and similar questions have been modified according to the journal format and the required font format。

Question:Is there any individual difference in optimizing IPC intervention protocols?

Answer:Dear Reviewer,Thank you for your inquiry regarding individual differences in optimizing IPC intervention protocols. You are correct in noting that individual variability is a critical consideration in sports science research, including studies on IPC. While it is true that individual differences do exist and can influence the outcomes of IPC interventions, our study implemented several experimental controls to minimize their impact and ensure the reliability of our findings.

Experimental Controls Implemented:

1)Homogeneous Participant Selection: We carefully selected a group of participants with similar training backgrounds and experience levels. This helped to control for variability in baseline strength and conditioning, which could otherwise confound the effects of IPC interventions.

2)Standardized Protocols: All participants followed identical testing and intervention protocols. This standardization ensured that any observed differences in outcomes could be more confidently attributed to the IPC interventions rather than variations in how the exercises were performed.

3)Double-Blind Design: The study was conducted in a double-blind manner, where neither the participants nor the researchers knew which intervention was being applied. This approach helps to eliminate bias that could arise from expectations or knowledge of the intervention.

4)Controlled Environment: The testing environment was controlled for variables such as temperature, time of day, and equipment used. This consistency reduces environmental factors that could lead to variability in performance.

5)Baseline Measurements: Prior to the intervention, all participants underwent baseline measurements to assess their initial strength levels. This data served as a reference point, allowing for individualized analysis and control.

6)Randomization: The order of IPC interventions was randomized, which helps to distribute any potential confounding effects evenly across all conditions.

While we acknowledge that individual differences cannot be entirely eliminated, the controls mentioned above were put in place to ensure that the study's findings are as robust and generalizable as possible. We also collected and analyzed data on various individual characteristics (e.g., training experience, anthropometric measures) to assess their influence on the outcomes. This approach allowed us to account for individual variability in our statistical analyses.In conclusion, although individual differences are inherent in any human study, the rigorous experimental design and controls employed in our study help to mitigate their impact and strengthen the validity of our conclusions regarding the effects of IPC interventions.

We hope this response addresses your concern and provides a clear explanation of how we managed individual variability in our research.

Question:If there is a relationship between the feature of physiological responses and the optimal duration of IPC intervention, the contribution of this paper to the sensing technologies and applications becomes clear, and it will be helpful for determining the duration for each person.

Answer:Dear Reviewer, thank you for your insightful question regarding the relationship between physiological response characteristics and the optimal duration of IPC intervention. Your suggestion to explore this relationship is both pertinent and valuable.Our study indeed aimed to investigate whether there is a correlation between individual physiological responses and the effectiveness of different IPC durations. The potential to tailor IPC protocols to individual physiological profiles could significantly enhance the efficacy of training and recovery interventions.

Contribution to Sensing Technologies and Applications:

1)Personalized Training Protocols: By understanding the relationship between physiological responses and IPC intervention duration, we can contribute to the development of personalized training protocols. This is where advanced sensing technologies play a crucial role, as they provide the data necessary to customize these protocols.

2)Optimizing IPC Duration: Our findings could help determine the optimal IPC duration for individuals, maximizing the benefits of IPC while minimizing potential risks or unnecessary strain. This is particularly important in high-performance sports, where the balance between training intensity and recovery is critical.

3)Data-Driven Decision Making: The use of advanced sensors allows for data-driven decision making in sports training. By monitoring physiological responses in real-time, coaches and athletes can make informed decisions about training loads and recovery strategies.

Future Research Directions: We agree with your suggestion and plan to explore this relationship further in future research. By collecting more granular data on physiological responses and correlating it with the effects of IPC interventions, we can better understand how to optimize the duration of IPC for each individual.In conclusion, the relationship between physiological response characteristics and the optimal IPC duration is a fascinating area of research with practical implications for sports performance and recovery. We are committed to advancing this field and appreciate your valuable input.

We tried our best to improve the manuscript and made some changes marked in red in the revised paper, which will not influence the content and framework of the paper. We appreciate the Editors/Reviewers’ diligent work and hope the corrections meet with approval. Thank you very much for your comments and suggestions once again.

Best wishes!

Thank you once again for your thorough review!

Xuehan Niu  Qifei Xia  Jie Xu   Li Tang

Reviewer 3 Report

Comments and Suggestions for Authors

REVIEW FOR MANUSCRIPT ENTITLED

“ASSESSMENT OF THE IMPACT OF SENSOR-BASED ISCHEMIC PRECONDITIONING WITH DIFFERENT CYCLING PERIODS ON UPPER LIMB STRENGTH IN BODYBUILDING ATHLETES

Thank you for providing me with the opportunity to review the manuscript entitled “ASSESSMENT OF THE IMPACT OF SENSOR-BASED ISCHEMIC PRECONDITIONING WITH DIFFERENT CYCLING PERIODS ON UPPER LIMB STRENGTH IN BODYBUILDING ATHLETES”.

Congratulations on the important work done. It can be seen that a significant effort has been made in the research included in this paper.

It is an interesting work of high methodological quality. Correct definition of terms, perfect analysis, explicit definition of the methods used to prepare the paper, etc. There is a correct structure, grammar and basic aspects in the publication.

However, there are certain changes that should be made. These are small modifications that should be made, such as:

Since cardiovascular diseases are mentioned in the introduction, it would be interesting to note that they are one of the main causes of pathology in the world and that ischemic preconditioning is used as a medical procedure.   In this way, the subject of the study would be further introduced.

We talk about cardiovascular diseases but we could specify that it is a sudden event such as myocardial damage, a sudden and fortuitous event such as acute myocardial infarction. These are congestive type diseases but it should be clear in the introduction and not only in the inclusion criteria.

If so, can it be applied to any type of patient? Low risk patients? High risk patients?

Have they received post PCI procedure cardiac rehabilitation?

The objective of the study should be more clearly specified in order to give a specific answer in the conclusion.

2. Objective and methods

Very interesting to emphasize this sentence: “The decision to select male participants was to control for hormonal and physiological variations that could impact muscle activation and training outcomes, ensuring consistency in physiological measurements”.

It is perfect to note that they have been informed and collected the informed consents and passed ethics committees.

Correctly described the method.

The Experimental flow chart, the images and the tables are very explanatory.

3. Results

All correctly described.

4, Discussion

A great deal of work has been done in discussing the subject. All key aspects have been touched upon.

5. Conclusion

Perfectly synthesized the final result.

It is very important to highlight the high quality of the work and all the effort made. It is appreciated that it is a great work but certain aspects still need to be modified for the quality of the publication.

After these recommendations, I hope they will be useful and helpful in your publication.

Author Response

Reviewer 3-----Reply

Dear reviewer 3:

We feel great thanks for your professional review work on our article. As you are concerned, there are several problems that need to be addressed. According to your nice suggestions, we have made extensive corrections to our previous draft, the detailed corrections are listed below.

Question:Since cardiovascular diseases are mentioned in the introduction, it would be interesting to note that they are one of the main causes of pathology in the world and that ischemic preconditioning is used as a medical procedure.   In this way, the subject of the study would be further introduced.

We talk about cardiovascular diseases but we could specify that it is a sudden event such as myocardial damage, a sudden and fortuitous event such as acute myocardial infarction. These are congestive type diseases but it should be clear in the introduction and not only in the inclusion criteria.

If so, can it be applied to any type of patient? Low risk patients? High risk patients?

Have they received post PCI procedure cardiac rehabilitation?

The objective of the study should be more clearly specified in order to give a specific answer in the conclusion.

Answer:Dear Reviewer, thank you for your valuable feedback on the relevance of IPC in the context of cardiovascular diseases and its application in medical procedures. Your suggestions to clarify these aspects in the introduction and to specify the study's objectives are well taken.We will revise the introduction to provide a clearer link between the medical application of IPC in cardiovascular diseases and its adaptation in sports science. This will help set the stage for our research and highlight the significance of IPC beyond its traditional medical context.

Modified Introduction:

Ischemic preconditioning (IPC), a procedure originally developed for the management of cardiovascular diseases, has been recognized for its potential in conditioning the heart against ischemic events. As one of the leading causes of global pathology, cardiovascular diseases, including acute myocardial infarction, present a sudden and often fortuitous challenge to heart health. The application of IPC as a preventive measure in these scenarios is well-established.

In addition, we clarification on patient types and rehabilitation: Our study primarily focuses on athletic populations; however, your query prompts us to consider the broader implications of IPC. While our current research does not directly involve patient groups, the principles we investigate could potentially be extrapolated to various patient profiles. Future studies could indeed explore the application of IPC in different risk categories, including low and high-risk patients, and its role in post-PCI cardiac rehabilitation.

We appreciate your insightful questions and suggestions, which have helped us to further strengthen the relevance and clarity of our research.

  1. Objective and methods

Very interesting to emphasize this sentence: “The decision to select male participants was to control for hormonal and physiological variations that could impact muscle activation and training outcomes, ensuring consistency in physiological measurements”.

It is perfect to note that they have been informed and collected the informed consents and passed ethics committees.

Correctly described the method.

The Experimental flow chart, the images and the tables are very explanatory.

  1. Results

All correctly described.

4, Discussion

A great deal of work has been done in discussing the subject. All key aspects have been touched upon.

  1. Conclusion

Perfectly synthesized the final result.

It is very important to highlight the high quality of the work and all the effort made. It is appreciated that it is a great work but certain aspects still need to be modified for the quality of the publication.

We tried our best to improve the manuscript and made some changes marked in red in the revised paper, which will not influence the content and framework of the paper. We appreciate the Editors/Reviewers’ diligent work and hope the corrections meet with approval. Thank you very much for your comments and suggestions once again.

Best wishes!

Thank you once again for your thorough review!

Xuehan Niu  Qifei Xia  Jie Xu   Li Tang

Round 2

Reviewer 1 Report

Comments and Suggestions for Authors

The authors improved the article according to the reviewers' suggestions.